# *Thalassophryne maculosa* (Batrachoididae: Thalassophryninae) as a Bioindicator of Mercury-Induced Genotoxicity

**DOI:** 10.3390/toxics13030206

**Published:** 2025-03-13

**Authors:** Mauro Nirchio Tursellino, Nicola Noemi Coppola, Juan Ignacio Gaviria Montoya, Juan Antonio Gómez

**Affiliations:** 1Departamento de Acuicultura, Facultad de Ciencias Agropecuarias, Universidad Técnica de Machala, Av. Panamericana km 5.5, Vía Pasaje, Machala 070150, Ecuador; 2Departamento de Acuicultura, Escuela de Ciencias Aplicadas del Mar, Universidad de Oriente, Boca de Río 6301, Estado Nueva Esparta, Venezuela; nicolanoemicoppola@gmail.com (N.N.C.); gaviria.ji@gmail.com (J.I.G.M.); 3Facultad de Ciencias Naturales, Exactas y Tecnología, Universidad de Panamá, Panamá 3366, Panama; juanay05@hotmail.com

**Keywords:** bioindicator, genotoxicity, marine pollution, mercury, micronuclei, *Thalassophryne maculosa*

## Abstract

Environmental monitoring requires reliable bioindicators to assess the genotoxic effects of pollutants in aquatic ecosystems. In this study, the marine fish *Thalassophryne maculosa* was evaluated as a bioindicator of genotoxicity through the application of the micronucleus test. Fish were exposed to varying concentrations of mercuric chloride (HgCl_2_) (0.1, 0.25, and 0.5 µg HgCl_2_/g body weight) over different time intervals (24, 48, 72, and 96 h). A dose- and time-dependent increase in nuclear abnormalities, including micronuclei, was observed, with significant chromosomal damage detected at 0.25 and 0.5 µg HgCl_2_/g body weight. These results demonstrate the sensitivity of *T. maculosa* to mercury exposure, even at concentrations below regulatory safety thresholds, emphasizing its suitability as a bioindicator for detecting genotoxic contamination in coastal ecosystems. This study provides critical insights into the ecological risks posed by mercury and highlights the potential of *T. maculosa* to enhance environmental monitoring programs, particularly in regions vulnerable to heavy metal pollution.

## 1. Introduction

Human activities such as deforestation, agriculture, construction, and industrial expansion have drastically elevated pollutant emissions, overwhelming ecosystems’ natural ability to mitigate them. This increase in contamination has had profound consequences, from cellular disruptions to ecosystem-wide impacts on biodiversity and human health [1,2,3,4,5,6]. Among these pollutants, heavy metals such as mercury stand out for their extreme toxicity, environmental persistence, and capacity for bioaccumulation and biomagnification in aquatic food webs, posing serious risks to marine and freshwater ecosystems [7,8,9,10,11].

In aquatic ecosystems, mercury occurs in three main forms: elemental (Hg^0^), inorganic (Hg^2+^) and organic (methylmercury, MeHg). While Hg^0^ is relatively inert, Hg^2+^ and MeHg are the most toxic due to their strong biological impact [12]. Through microbial methylation, Hg^2+^ is converted into MeHg, the most dangerous form, as it bioaccumulates in fish tissues and biomagnifies along the food web. As a result, top predators face the highest exposure, with MeHg concentrations up to six orders of magnitude higher than those present in surface water [13]. This prolonged accumulation generates oxidative stress, alters enzymatic defenses, and causes physiological and neurological dysfunctions, affecting the health of exposed organisms. In addition, MeHg accumulates predominantly in muscle tissues, where it accounts for more than 90% of total mercury, increasing its toxicity to aquatic predators [14].

Beyond oxidative stress and enzymatic alterations, mercury exposure has been associated with genotoxic effects, such as chromosomal instability and micronucleus formation, which compromise DNA integrity [15,16]. Likewise, histopathological alterations, such as hyperplasia, inflammation, necrosis, and nuclear abnormalities, reinforce the magnitude of the ecological impact of mercury contamination in aquatic ecosystems [17].

Mercury contamination in aquatic ecosystems, especially near industrial discharge, agricultural runoff and sewage outfalls, represents a serious threat to water quality and marine biodiversity [1,18,19,20]. Fish, as an integral part of the food web, play a key role in environmental assessment due to their ability to bioaccumulate toxins and respond to low concentrations of pollutants, making them ideal bioindicators of the ecological impact of pollution [21,22].

The micronucleus assay is a widely used tool for detecting genotoxicity. Originally developed to assess chromosomal damage in rodents [23,24,25], this method identifies extranuclear bodies formed by chromosomal fragments or complete chromosomes that fail to integrate into daughter nuclei during cell division [26,27]. These micronuclei are reliable indicators of genotoxicity and chromosomal instability. Because of its sensitivity to environmental pollutants, it has been adapted to various organisms, most notably in fish, where it is used to assess the impact of pollution on aquatic ecosystems [22,28,29,30].

Numerous studies have validated the use of fish as bioindicators for genotoxicity caused by contaminants, with the micronucleus assay as one of the most widely used tools. In marine environments, species such as *Odontesthes argentinensis*, *Paralichthys orbignyanus*, *Micropogonias furnieri*, and *Mugil platanus* have demonstrated sensitivity to genotoxic agents [31]. Similarly, *Epinephelus chlorostigma*, *Scomberomorus commerson* [32], *Limanda limanda*, and *Melanogrammus aeglefinus* [33] have been used in marine pollution studies. Freshwater species like *Andinoacara rivulatus* [15], *Danio reiro* [34,35], *Cyprinus carpio*, *Astyanax eigenmanniorum*, *Cheirodon interruptus* [36], *Oreochromis niloticus* [37,38,39], *Labeo rohita* [40], *Prochilodus magdalenae*, *Hoplias malabaricus* [41], and *Collosoma macropomum* [42], among others, have been effective in detecting genotoxicity with this assay.

While most studies have focused on commercially important freshwater and marine species, estuarine benthic fish have been explored little in terms of their genotoxic response to mercury, despite inhabiting ecosystems subject to environmental fluctuations and variable levels of contamination.

Since metabolic and ecological characteristics influence sensitivity to contaminants, it is essential to extend research toward new bioindicator organisms. In this context, *Thalassophryne maculosa* Günther, 1861, known as ‘cano toadfish’, emerges as a relevant model for assessing the effects of heavy metal pollution. This venomous fish of the family Batrachoididae inhabits marine and brackish waters along the northern coast of South America and surrounding islands [43]. Although it lacks commercial value, it has attracted scientific interest due to its distinctive camouflage, sound production, cytogenetic characteristics and ability to survive in estuarine environments, ecosystems particularly vulnerable to heavy metal pollution [44,45,46]. These characteristics make it an ecologically relevant model for assessing mercury-induced genotoxicity [47,48].

In this study, we applied the micronucleus assay to assess chromosomal damage in *T. maculosa* individuals exposed to varying concentrations of inorganic mercury. The goal was to establish the species’ potential as a bioindicator for mercury-induced genotoxicity and contribute to a deeper understanding of the ecological risks posed by heavy metal pollution in marine environments.

## 2. Materials and Methods

### 2.1. Specimen Collection and Acclimation

Fish specimens for this study were collected on the same day from the entrance of La Restinga Lagoon, Margarita Island, Venezuela (coordinates: 10°58.7′ N, 64°9.7′ W). Through free diving at a depth of 1.2 m, specimens were carefully captured with hand nets to minimize stress and potential injury. Immediately after collection, the fish were transported to the facilities of the Escuela de Ciencias Aplicadas del Mar (E.C.A.M) at the Universidad de Oriente.

Upon arrival, the fish were housed in a concrete holding tank equipped with aeration and a controlled flow of seawater to maintain water quality. They were acclimated under stable environmental conditions for 48 h to allow recovery and adjustment prior to the experimental procedures. During this period, no feeding was provided to standardize physiological conditions among individuals. Environmental parameters, such as temperature, salinity, dissolved oxygen, and pH, were monitored regularly to ensure consistency with the collection site.

### 2.2. Experimental Setup and Grouping

Following acclimatization, the fish were randomly assigned to 12 experimental groups, each comprising five individuals. The groups were housed in cages measuring 80 cm × 60 cm × 30 cm, constructed with plastic mesh featuring 1-inch openings to allow water flow. These cages were suspended 50 cm below the tidal level at the distal end of the E.C.A.M dock, approximately 65 m offshore, ensuring continuous exposure to high-quality seawater.

While the suspended setup did not fully replicate the benthic conditions typically inhabited by *Thalassophryne maculosa*, the plastic mesh provided a stable surface for the fish to rest and reduced potential stress by maintaining proximity to their natural aquatic environment. To account for potential mortality, each group initially contained five fish; however, all individuals survived during the study. For subsequent analyses, three fish were randomly selected from each group to ensure consistency. To maintain uniform experimental conditions and avoid confounding variables, the fish were not fed throughout the duration of the study.

### 2.3. Factorial Design and Treatment Administration

A factorial design was employed to evaluate the effects of HgCl_2_ concentration and exposure duration on the test specimens. Fish were exposed to four concentrations of mercuric chloride (HgCl_2_)—0.0, 0.1, 0.25, and 0.5 µg/g body wet weight—with the control group (0 µg/mL) receiving only distilled water. The exposure levels were achieved through intraperitoneal injection at a dose of 1 mL per 100 g of fish weight, ensuring precise and uniform administration while minimizing variability due to differential uptake or metabolic differences.

The exposure duration included four time points (24, 48, 72, and 96 h) to capture both acute and progressive genotoxic effects. This factorial approach allowed for the independent and interactive assessment of HgCl_2_ concentration and exposure duration, providing a comprehensive evaluation of its genotoxic potential.

### 2.4. Selection of HgCl_2_ as the Experimental Compound

Mercury(II) chloride (HgCl_2_) was chosen for its high solubility in water and relevance to aquatic environments. Inorganic mercury (Hg^2+^)—including HgCl_2_—is a predominant form of mercury pollution introduced into aquatic ecosystems through industrial and anthropogenic activities [49,50]. Once released into the environment, HgCl_2_ remains highly bioavailable and can undergo microbial methylation, transforming into methylmercury and contributing to mercury cycling [12].

Additionally, HgCl_2_ is widely used in toxicological studies as a representative form of inorganic mercury due to its strong interaction with biological systems. Its use facilitates comparisons with the existing literature on mercury toxicity in fish species [49,50], making it a suitable model for assessing genotoxic effects in aquatic organisms.

### 2.5. Blood Sampling and Micronucleus Analysis

Blood samples were collected at 24, 48, 72, and 96 h of exposure from fish in each treatment group. Sampling was performed via caudal vein puncture using a heparinized syringe to prevent coagulation. A small volume of blood was spread evenly onto two clean glass slides per fish to ensure sufficient sample representation. The slides were left to air dry at room temperature, and then fixed in absolute methanol for 10 min to preserve cellular structures. Subsequently, the slides were stained with a 5% Giemsa solution (pH 6.8) for 10 min to enhance chromosomal and nuclear visibility.

To achieve reliable quantification, the number of photographs taken per slide was adjusted based on the concentration of erythrocytes in each sample, ensuring that at least 1000 cells per slide were evaluated. Digital images were captured at 1000× magnification using a Motic photomicroscope, with care taken to avoid overlapping or damaged cells during analysis.

Micronucleus scoring was conducted by a single, experienced observer following the standardized guidelines of Nirchio et al. [28] to minimize inter-observer variability. Micronuclei were recorded as the number observed per 2000 erythrocytes per individual, providing a precise estimate of micronucleus frequency and minimizing the impact of interindividual variability. This approach is consistent with previous studies in genotoxic biomonitoring in fish, where analyzing a high number of cells per individual compensates for the use of a moderate sample size, ensuring the detectability of significant differences between treatments. Furthermore, the selection of an optimized sample size adheres to the ethical principle of reduction in animal experimentation, in accordance with international guidelines for aquatic toxicology studies.

Other nuclear abnormalities were noted but not included in the primary micronucleus frequency analysis.

### 2.6. Statistical Analyses

Initial statistical evaluations revealed that the data did not meet the assumptions of normality (Shapiro–Wilk test) or homoscedasticity (Levene test). To address these violations, an inverse square root transformation [1/√(Micronuclei/2000 cells)] was applied, enabling the use of parametric analysis of variance (ANOVA). The transformed data were analyzed to assess the independent effects of mercuric chloride (HgCl_2_) concentration and exposure time, as well as their interaction. Post hoc comparisons were performed using the Least Significant Difference (LSD) test at a 95% confidence level, identifying statistically significant differences (*p* < 0.05) across treatments [51,52].

Statistical analyses were conducted using StatGraphics Centurion XVI software. Version 16.1.18. In order to provide a comprehensive visual representation overview of the dose- and time-dependent effects of HgCl_2_ on chromosomal damage, a three-dimensional plot was generated in Python (version 3.8) using the Matplotlib library (version 3.5.1) [53]. The plot_trisurf function was employed to create a 3D surface illustrating the relationship between HgCl_2_ concentration, exposure time, and micronucleated cell frequency. The ‘viridis’ colormap was applied, with lighter tones representing higher frequencies of micronuclei.

## 3. Results

Salinity in the collection area ranged between 36.1 and 37.8 g/L, with water temperature between 21.8 and 27.4 °C, pH between 7.4 and 7.8, and dissolved oxygen concentrations between 4.9 and 7.3 mg/L. Experimental conditions remained stable, with salinity (35.6–38.0 g/L), temperature (22.1–28.5 °C), pH (7.3–7.7), and dissolved oxygen (5.5–7.0 mg/L) falling within expected ranges.

Microscopic analysis of blood smears revealed a variety of abnormalities, including micronucleated mature erythrocytes (MMEs), micronucleated immature erythrocytes (MIEs), binucleated immature erythrocytes (BIEs), binucleated immature erythrocytes with a cytoplasmic bridge (BIECBs), and erythrocytes exhibiting vacuoles or cytoplasmic loss (VLCs) (Figure 1). Normal erythrocytes were characterized by an oval or ellipsoidal shape, an elliptical nucleus, and cytoplasm free of nuclear aggregates

The ANOVA results (Table 1) demonstrate that both HgCl_2_ concentration and exposure time had highly significant effects on micronucleus frequency (*p* < 0.01). The interaction between these two factors, however, was not significant (F = 0.26, *p* = 0.9819), indicating that their effects were additive rather than synergistic.

Post hoc analysis revealed that the frequency of micronucleated cells significantly increased with higher HgCl_2_ concentrations and longer exposure times (Table 2). Treatments were categorized into homogeneous subsets, confirming distinct thresholds for genotoxic effects.

The micronucleus frequency showed a clear dose- and time-dependent pattern (Figure 2). At lower concentrations (0 and 0.1 µg/g body weight), micronucleated cells remained relatively stable over time, with a frequency ranging between 1 and 5 MN/2000 cells. This indicates minimal genotoxicity at these exposure levels. In contrast, higher concentrations (0.25 and 0.5 µg/g body weight) led to a pronounced increase in micronucleated cells, particularly with extended exposure.

Exposure to 0.5 µg/g body weight for 96 h resulted in a mean frequency of up to 25 MN/2000 cells, highlighting the intensified genotoxic effects of mercury with both increasing concentration and prolonged exposure.

## 4. Discussion

The dose- and time-dependent increase in nuclear abnormalities observed in *Thalassophryne maculosa* reinforces the well-documented genotoxic potential of mercury in aquatic organisms [15,54,55,56,57].

Exposure to higher concentrations of HgCl_2_ significantly elevated the micronucleus frequency, providing strong evidence of mercury’s ability to induce chromosomal damage in fish erythrocytes. The low baseline frequency of micronucleated cells in the control group aligns with spontaneous micronuclei levels reported in other fish species, such as *Astyanax bimaculatus*, *Colossoma macropomum*, and *Oreochromis niloticus* [39,42,58].

To ensure ecological relevance, this study was conducted in situ using cages placed in the sea. This setup allowed fish to be exposed to a natural environment while ensuring controlled mercury exposure conditions. Throughout the study, key water parameters—including salinity, temperature, dissolved oxygen, and pH—remained stable, minimizing potential confounding effects. Although conditions remained stable during the experiment, seasonal variations in salinity and temperature may influence mercury bioaccumulation and toxicity under natural conditions. Future studies should incorporate these variables to better understand their role in *T. maculosa*’s response to mercury contamination.

*T. maculosa* exhibited heightened sensitivity to HgCl_2_, showing significant increases in micronuclei, even at concentrations below regulatory safety thresholds. The tested exposure levels (0.1–0.5 µg/g) represent environmentally relevant contamination scenarios, with the highest concentration corresponding to the maximum allowable mercury level in fish and seafood for human consumption [59]. These findings suggest that *T. maculosa* is a promising bioindicator for the early detection of mercury contamination in estuarine and coastal environments.

The genotoxic effects observed in *T. maculosa* align with findings in other teleost species, including *Danio rerio*, *O. niloticus*, and *Prochilodus magdalenae*, where similar dose- and time-dependent increases in nuclear abnormalities have been reported [39,60]. However, the higher micronucleus frequencies and nuclear abnormalities detected at lower HgCl_2_ concentrations in *T. maculosa* suggest greater sensitivity compared to some commercially important species, such as *O. niloticus* and *Cyprinus carpio* [61]. This variability in genotoxic responses may be attributed to differences in DNA repair efficiency, antioxidant defense mechanisms, and metabolic adaptations. Understanding these interspecific differences is crucial for selecting species in mercury contamination monitoring programs. A distinct threshold for genotoxicity was identified in *T. maculosa*, with minimal effects at 0.1 µg/g but a significant increase in micronucleated cells at 0.25–0.5 µg/g. This pattern is consistent with findings in *P. magdalenae* and *O. niloticus*, which also exhibit dose-dependent responses but varying species-specific sensitivities [39,60]. Notably, statistical analysis confirmed that the effects of HgCl_2_ concentration and exposure time were additive rather than synergistic, mirroring findings in *Clarias gariepinus* [32].

The genotoxic effects observed in *T. maculosa* are consistent with mercury-induced oxidative stress and chromosomal instability. In species like *C. carpio*, HgCl_2_ inhibits key antioxidant enzymes such as superoxide dismutase (SOD) and catalase (CAT), leading to reactive oxygen species (ROS) accumulation and cellular damage [61]. While oxidative stress likely plays a role in T. maculosa, biochemical markers such as antioxidant enzyme activity or lipid peroxidation were not assessed in this study. Future research should incorporate these biomarkers to clarify mercury’s mechanistic effects in this species.

Additionally, the formation of micronuclei and other nuclear abnormalities may be linked to clastogenic and aneugenic mechanisms. Mercury’s strong affinity for sulfhydryl groups in proteins can interfere with DNA repair pathways and mitotic spindle function, leading to chromosomal instability and cell cycle errors [61,62]. However, additional cytogenetic assays are needed to differentiate between these mechanisms in *T. maculosa*.

Overall, these findings establish *T. maculosa* as a highly sensitive species for detecting genotoxic contamination in coastal ecosystems. Its ability to detect mercury-induced genotoxicity, even at low concentrations, underscores its potential as a valuable tool for environmental monitoring and risk assessment.

## Figures and Tables

**Figure 1 toxics-13-00206-f001:**
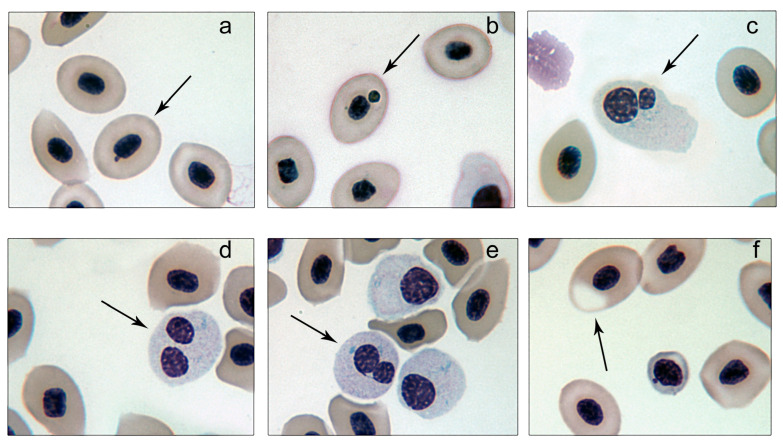
Nuclear abnormalities in peripheral blood of *Thalassophryne maculosa*. (**a**,**b**) Mature micronucleated erythrocytes (MMEs) with micronuclei of varying sizes; (**c**) immature binucleated erythrocytes (BIEs); (**d**) mature binucleated erythrocytes (MBEs); (**e**) immature binucleated erythrocytes with a cytoplasmic bridge (BIECBs); (**f**) erythrocytes with vacuoles or cytoplasmic loss (VLCs).

**Figure 2 toxics-13-00206-f002:**
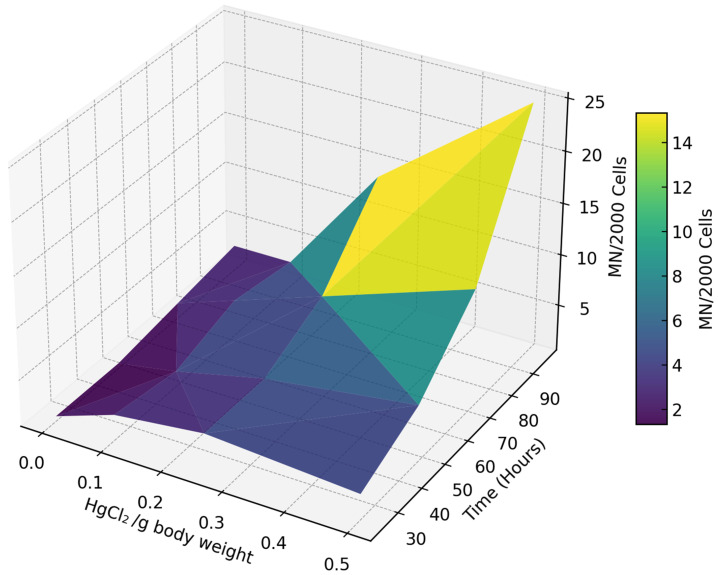
A three-dimensional representation of the variation in micronucleated cells (MN/2000 cells) in response to HgCl_2_ concentration (µg/body weight) and exposure time (hours). Darker colors (blue/green) indicate fewer micronucleated cells, while lighter colors (yellow) denote higher frequencies.

**Table 1 toxics-13-00206-t001:** Analysis of variance for 1/(√(Micronuclei/2000 cells^2^)).

Source	Sum of Squares	df	Mean Square	F-Ratio	*p*-Value
Main Effects					
A: HgCl_2_ (µg/g body weight)	0.71824	3	0.23941	21.17	0.0000
B: Exposure Time (hours)	0.66489	3	0.22163	19.59	0.0000
Interactions					
AB	0.02609	9	0.00290	0.26	0.9819
Residuals	0.36194	32	0.01131		
Total	1.77117	47			

**Table 2 toxics-13-00206-t002:** Post hoc LSD test results for the effects of HgCl_2_ concentration and exposure time on micronucleus frequency in *Thalassophryne maculosa.*

Variable	Level	Mean LS	Homogeneous Groups
Concentration	0.50 µg/g	0.3876	A
Concentration	0.25 µg/g	0.4140	A
Concentration	0.10 µg/g	0.6277	B
Concentration	0.00 µg/g	0.6597	B
Exposure Time	96 h	0.3659	A
Exposure Time	72 h	0.4549	A
Exposure Time	48 h	0.6072	B
Exposure Time	24 h	0.6609	B

## Data Availability

The data generated in this study are not publicly available but can be obtained upon reasonable request from the corresponding author.

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
