# Peer review of "Thalassophryne maculosa (Batrachoididae: Thalassophryninae) as a Bioindicator of Mercury-Induced Genotoxicity"

_toxics, 2025, doi:10.3390/toxics13030206_

Round 1

Reviewer 1 Report

Comments and Suggestions for Authors

Review Report: toxics-3505250-peer-review-v1

Dear Authors,

The manuscript titled "Thalassophryne maculosa (Batrachoididae: Thalassophryninae) as a Bioindicator of Mercury-Induced Genotoxicity" presents compelling evidence supporting the use of T. maculosa as a sensitive bioindicator in coastal ecosystems affected by heavy metal pollution. The study employs the micronucleus test to assess the genotoxic effects of mercuric chloride (HgCl₂) exposure across different concentrations (0.1, 0.25, and 0.5 μg HgCl₂/g body weight) and time intervals (24, 48, 72, and 96 hours). The findings indicate a clear dose- and time-dependent increase in chromosomal damage, with significant effects detected at concentrations as low as 0.25 μg HgCl₂/g body weight. These results highlight the acute sensitivity of T. maculosa to mercury contamination, reinforcing its potential as a bioindicator for environmental monitoring.

Major Comments and Suggestions for Improvement

While the study provides valuable insights into mercury-induced genotoxicity, several aspects require further clarification to enhance its scientific contribution:

  1. Clarification of Novelty

Although the findings align with existing knowledge on mercury’s genotoxic effects in aquatic organisms, the manuscript does not clearly articulate its novel contributions. The dose- and time-dependent increase in nuclear abnormalities has been well-documented in other fish species. Therefore, the authors should explicitly discuss whether T. maculosa exhibits unique sensitivities or mechanistic differences compared to established bioindicator species. Addressing this point would better position the study within the broader scientific literature.

  1. Contextualization within Existing Research

The manuscript would benefit from a clearer discussion of how these results compare with previous studies on mercury-induced genotoxicity. Have similar exposure concentrations and effects been observed in other fish species? Does T. maculosa exhibit a heightened sensitivity relative to commercially important species? Providing such comparisons would strengthen the ecological relevance of the study.

  1. Mechanistic Insights into Genotoxicity

The discussion attributes genotoxic effects to oxidative stress and chromosomal instability but lacks direct biochemical evidence to support this claim. Have the authors considered measuring oxidative stress biomarkers, such as antioxidant enzyme activity (e.g., superoxide dismutase, catalase) or lipid peroxidation levels? While not essential for the current study, acknowledging the need for such analyses in future research would provide a more comprehensive understanding of mercury’s mode of action.

  1. Environmental and Regulatory Implications

Given that the tested mercury concentrations correspond to regulatory safety thresholds for seafood consumption, a discussion on how T. maculosa compares to other species in terms of sensitivity to mercury exposure would be valuable. Would its high sensitivity make it a particularly effective bioindicator for early detection of contamination? Additionally, integrating field data alongside laboratory results could enhance the ecological validity of the findings.

  1. Potential for Long-Term and Environmental Variability Studies

To further improve the study, I would like to ask the authors whether they have considered the following:

  • Have variations in environmental conditions, such as salinity and water temperature, been explored in relation to mercury bioaccumulation and genotoxicity? If not, do the authors believe these factors could influence maculosa’s response to mercury exposure?
  • Have any long-term exposure experiments been conducted to assess potential adaptive or chronic effects of mercury contamination? If not, how might prolonged exposure impact the species' physiological or genetic response to heavy metal toxicity?
  • Have the authors considered integrating field studies alongside controlled laboratory experiments to validate the ecological relevance of their findings?

Recommendation

Based on the strengths of the study and the suggested revisions, I recommend major revisions to improve the clarity of its novelty, contextualize the findings within existing research, and expand the discussion on ecological implications. Addressing these aspects will significantly enhance the manuscript’s scientific contribution and impact.

Best regards,

Comments on the Quality of English Language

The English could be improved to more clearly express the research.

Author Response

Comments 1: Clarification of Novelty
Although the findings align with existing knowledge on mercury’s genotoxic effects in aquatic organisms, the manuscript does not clearly articulate its novel contributions. The dose- and time-dependent increase in nuclear abnormalities has been well-documented in other fish species. Therefore, the authors should explicitly discuss whether T. maculosa exhibits unique sensitivities or mechanistic differences compared to established bioindicator species. Addressing this point would better position the study within the broader scientific literature.

Response 1: Thank you for your insightful comment. We acknowledge the importance of clearly defining the novel contributions of our study. To address this, we have revised the Introduction (lines 80–89) to better position Thalassophryne maculosa within the context of mercury-induced genotoxicity research and to highlight the gap in knowledge regarding estuarine benthic species as bioindicators. Additionally, we have expanded the Discussion (lines 264–279) to explicitly compare the genotoxic response of T. maculosa to that of established bioindicator species, such as O. niloticus and C. carpio. We now emphasize that T. maculosa exhibited significant genotoxic effects at lower mercury concentrations than these species, suggesting a heightened sensitivity that may be linked to species-specific metabolic traits, detoxification pathways, or ecological niche adaptations. Furthermore, we discuss the implications of these findings for bioindicator selection, reinforcing the relevance of T. maculosa in estuarine environments, where mercury contamination often fluctuates. These clarifications strengthen the novelty of our study and its contribution to the broader field of ecotoxicology.

Comments 2: Contextualization within Existing Research
The manuscript would benefit from a clearer discussion of how these results compare with previous studies on mercury-induced genotoxicity. Have similar exposure concentrations and effects been observed in other fish species? Does T. maculosa exhibit a heightened sensitivity relative to commercially important species? Providing such comparisons would strengthen the ecological relevance of the study.

Response 2: Thank you for this valuable suggestion. We have expanded the Discussion section (lines 264-269) to provide a more detailed comparison of our findings with previous studies on mercury-induced genotoxicity in fish.

Comments 3: Mechanistic Insights into Genotoxicity
The discussion attributes genotoxic effects to oxidative stress and chromosomal instability but lacks direct biochemical evidence to support this claim. Have the authors considered measuring oxidative stress biomarkers, such as antioxidant enzyme activity (e.g., superoxide dismutase, catalase) or lipid peroxidation levels? While not essential for the current study, acknowledging the need for such analyses in future research would provide a more comprehensive understanding of mercury’s mode of action.

Response 3:
Thank you for this valuable suggestion. We acknowledge that direct biochemical evidence of oxidative stress was not assessed in this study. To address this, we have revised the Discussion section (lines 281-287) to explicitly recognize the need for future research incorporating oxidative stress biomarkers.

Comments 4: Environmental and Regulatory Implications
Given that the tested mercury concentrations correspond to regulatory safety thresholds for seafood consumption, a discussion on how T. maculosa compares to other species in terms of sensitivity to mercury exposure would be valuable. Would its high sensitivity make it a particularly effective bioindicator for early detection of contamination? Additionally, integrating field data alongside laboratory results could enhance the ecological validity of the findings.

Response 4: Thank you for your insightful comment. We have revised the Discussion section to further elaborate on T. maculosa's sensitivity to mercury and its potential role as a bioindicator species, particularly in the following parts:

  1. Lines referring to heightened sensitivity and its implications as a bioindicator (256-263). This section directly supports the statement that T. maculosa shows a genotoxic response even at regulatory safety limits, reinforcing its value as an early warning species.

  2. Comparison with other species (O. niloticus, C. carpio, P. magdalenae)  (Lines 264-269). This paragraph explicitly compares T. maculosa with other species, addressing its higher sensitivity and justifying its relevance in environmental monitoring.

  3. Recognition of interspecific differences (lines 269-273) This passage explains possible reasons for T. maculosa's increased sensitivity, which aligns with the reviewer’s request for a discussion on its comparative response to mercury exposure.

Response 5: Thank you for your valuable comments and insightful questions. We recognize the importance of considering environmental variability and long-term exposure effects in assessing mercury genotoxicity. To address these points, we have clarified our experimental conditions and outlined directions for future research in the Discussion section (lines 246-254).

  1. Environmental Variability and Mercury Genotoxicity
    We acknowledge that environmental factors such as salinity, temperature, and oxygen levels can influence mercury bioaccumulation and toxicity. However, in this study, water parameters were carefully monitored, and conditions remained stable throughout the experiment (Materials and Methods, lines 106-109; Results, lines 191-195). While this stability minimized confounding factors, we recognize that in natural settings, fluctuations in these parameters may alter mercury uptake and toxicity. Future studies should incorporate seasonal variations and site-specific environmental differences to evaluate their potential influence on genotoxic responses in T. maculosa.

  2. Long-Term Exposure and Potential Adaptive Responses
    This study focused on short-term exposure (24–96 hours) to assess acute genotoxic effects. While our results demonstrate a clear dose- and time-dependent response, they do not account for potential adaptive mechanisms or chronic toxicity effects. Prolonged mercury exposure could lead to physiological and genetic adaptations, including alterations in antioxidant defense systems, DNA repair efficiency, or epigenetic modifications. Future research should explore long-term exposure scenarios to determine whether T. maculosa develops tolerance mechanisms or accumulates irreversible genetic damage over extended periods.

  3. Field Studies vs. Controlled Experiments
    Our study was conducted in situ using cages placed in the sea, ensuring exposure to a natural environment while maintaining controlled mercury concentrations. This approach provides ecological relevance compared to traditional laboratory experiments. 

Reviewer 2 Report

Comments and Suggestions for Authors

The article concerns Thalassophryne maculosa (Batrachoididae: Thalassophryninae) as a Bioindicator of Mercury-Induced Genotoxicity. In order to publish the manuscript should be improved.

1. In the introduction, please add a fragment regarding the toxic effects of mercury on aquatic organisms.

2. It is also necessary to identify which forms of mercury occur in waters. To which forms of mercury are aquatic organisms exposed.

3. There is no information on the temperature, salinity, pH of water, etc., from which areas the fish were collected. I also do not see information on the conditions in which the fish were stored after collection.

4. Please explain why the form of mercury HgCl2 was used in the study.

5. Please expand the discussion of the results. The topic has only been touched on. Or consider publishing the manuscript only as a communication.

Comments on the Quality of English Language

It should be improved.

Author Response

Comments 1: In the introduction, please add a fragment regarding the toxic effects of mercury on aquatic organisms.

Response 1: Thank you for pointing this out. We agree with this comment. Therefore, we have added a fragment discussing the toxic effects of mercury on aquatic organisms in the Introduction section. Specifically, we included information about mercury’s ability to induce oxidative stress, impair enzymatic functions, and cause DNA damage in fish and other aquatic organisms (lines 43-47)

Comments 2: It is also necessary to identify which forms of mercury occur in waters. To which forms of mercury are aquatic organisms exposed.

Response 2: Thank you for pointing this out. We agree with this comment. Therefore, we have added a section in the Introduction clarifying the different forms of mercury present in aquatic environments and their impact on aquatic organisms. Specifically, we describe how mercury exists in elemental (Hg⁰), inorganic (Hg²⁺), and organic (methylmercury, MeHg) forms, with MeHg being the most bioavailable and toxic due to its ability to bioaccumulate and biomagnify through the food chain. This addition can be found on lines 37-43 of the revised manuscript.

Coment 3: There is no information on the temperature, salinity, pH of water, etc., from which areas the fish were collected. I also do not see information on the conditions in which the fish were stored after collection.

Response 3: Thank you for your observation. We have now clarified the environmental conditions of both the collection site and the experimental zone in the manuscript. In the Materials and Methods section (lines 107-109), we now specify that salinity, temperature, pH, and dissolved oxygen levels were monitored throughout the study to ensure stable conditions. These parameters were recorded at both the collection site and the experimental area to minimize potential environmental stressors that could influence the results. Additionally, in the Results section (lines 191-195), we have included the specific values of these parameters in both locations.

Comment 4: Please explain why the form of mercury HgCl2 was used in the study.

Response 4: Thank you for your question. We have now clarified the rationale for using mercury(II) chloride (HgCl₂) in the Materials and Methods section (lines 137-147). This explanation has now been incorporated into the revised manuscript. We appreciate your insightful comment, which has helped us improve the clarity of our study design.

Comment 5: Please expand the discussion of the results. The topic has only been touched on. Or consider publishing the manuscript only as a communication.

Response 5: Thank you for your suggestion. We have expanded the Discussion section to provide a more thorough interpretation of our results. Specifically, we have: Enhanced comparisons with previous studies on mercury-induced genotoxicity in other teleost species (Danio rerio, Oreochromis niloticus, Prochilodus magdalenae), improving the contextualization of our findings. Further explored potential mechanisms, discussing oxidative stress, chromosomal instability, and possible interference with DNA repair pathways. Additionally, we acknowledge the need for future studies on biochemical markers (e.g., antioxidant enzyme activity, lipid peroxidation). Elaborated on environmental implications, emphasizing T. maculosa’s suitability as a sentinel species for mercury contamination and the need for field-based studies to validate laboratory results.

Round 2

Reviewer 1 Report

Comments and Suggestions for Authors

Review Report: toxics-3505250-peer-review-v2

Dear Authors,

I commend your thorough revisions and thoughtful incorporation of the suggested recommendations. The updated manuscript successfully enhances the clarity of its novel contributions, provides a stronger contextualization within existing research, and expands on the ecological and regulatory implications of the findings. These improvements significantly strengthen the scientific depth and impact of the study. Additionally, the refined discussion and methodological clarifications contribute to a more comprehensive understanding of Thalassophryne maculosa as a bioindicator of mercury-induced genotoxicity.

Given these enhancements, I find the manuscript to be well-structured, scientifically sound, and suitable for publication in its current form.

Best regards.